# EasySpec: Layer-Parallel Speculative Decoding for Efficient Multi-GPU Utilization

**Yize Wu**[1,2]     **Ke Gao**[1]     **Ling Li**[1,2]     **Yanjun Wu**[1*]
[1]Intelligent Software Research Center, Institute of Software, CAS, Beijing, China
[2]University of Chinese Academy of Sciences, Beijing, China
{wuyize2021,gaoke,liling,yanjun}@iscas.ac.cn

## Abstract

Speculative decoding is an effective and lossless method for Large Language Model (LLM) inference acceleration. It employs a smaller model to generate a draft token sequence, which is then verified by the original base model. In multi-GPU systems, inference latency can be further reduced through tensor parallelism (TP), while the optimal TP size of the draft model is typically smaller than that of the base model, leading to GPU idling during the drafting stage. We observe that such inefficiency stems from the sequential execution of layers, which is seemingly natural but actually unnecessary. Therefore, we propose EasySpec, a layer-parallel speculation strategy that optimizes the efficiency of multi-GPU utilization. EasySpec breaks the inter-layer data dependency in the draft model, enabling multiple layers to run simultaneously across multiple devices as "fuzzy" speculation. After each drafting-and-verification iteration, the draft model's key-value cache is calibrated in a single forward pass, preventing long-term fuzzy-error accumulation at minimal additional latency. EasySpec is a training-free and plug-in method. We evaluated EasySpec on several mainstream open-source LLMs, using smaller versions of models from the same series as drafters. The results demonstrate that EasySpec can achieve a peak speedup of 4.17x compared to vanilla decoding, while preserving the original distributions of the base LLMs. Specifically, the drafting stage can be accelerated by up to 1.62x with a maximum speculation accuracy drop of only 7%. The code is available at https://github.com/Yize-Wu/EasySpec.

## 1   Introduction

Transformer-based Large Language Models (LLMs) have demonstrated remarkable problem-solving abilities across various domains [1–5]. However, as the parameter size continues to grow, the time-consuming process of auto-regressive decoding poses a significant barrier to deploying large models in latency-sensitive applications [6, 7].

Various effective approaches [8–10] have been proposed to reduce inference latency, including speculative decoding [11, 12] and tensor-parallel (TP) distribution [13]. Speculative decoding employs a smaller model to generate a draft token sequence, and uses token-level parallelism to conduct non-autoregressive verification, ensuring no shifting of the original model's output distribution. In contrast, TP distribution leverages cross-device parallelism by partitioning computational workloads across multiple devices (usually GPUs) and synchronizing the results subsequently, which is also lossless.

Combining speculative decoding with TP distribution achieves even greater acceleration ratio. However, integrating a draft model into a distributed system is not trivial [7]. Since the parameter size of

---

[*]Corresponding author.

39th Conference on Neural Information Processing Systems (NeurIPS 2025).

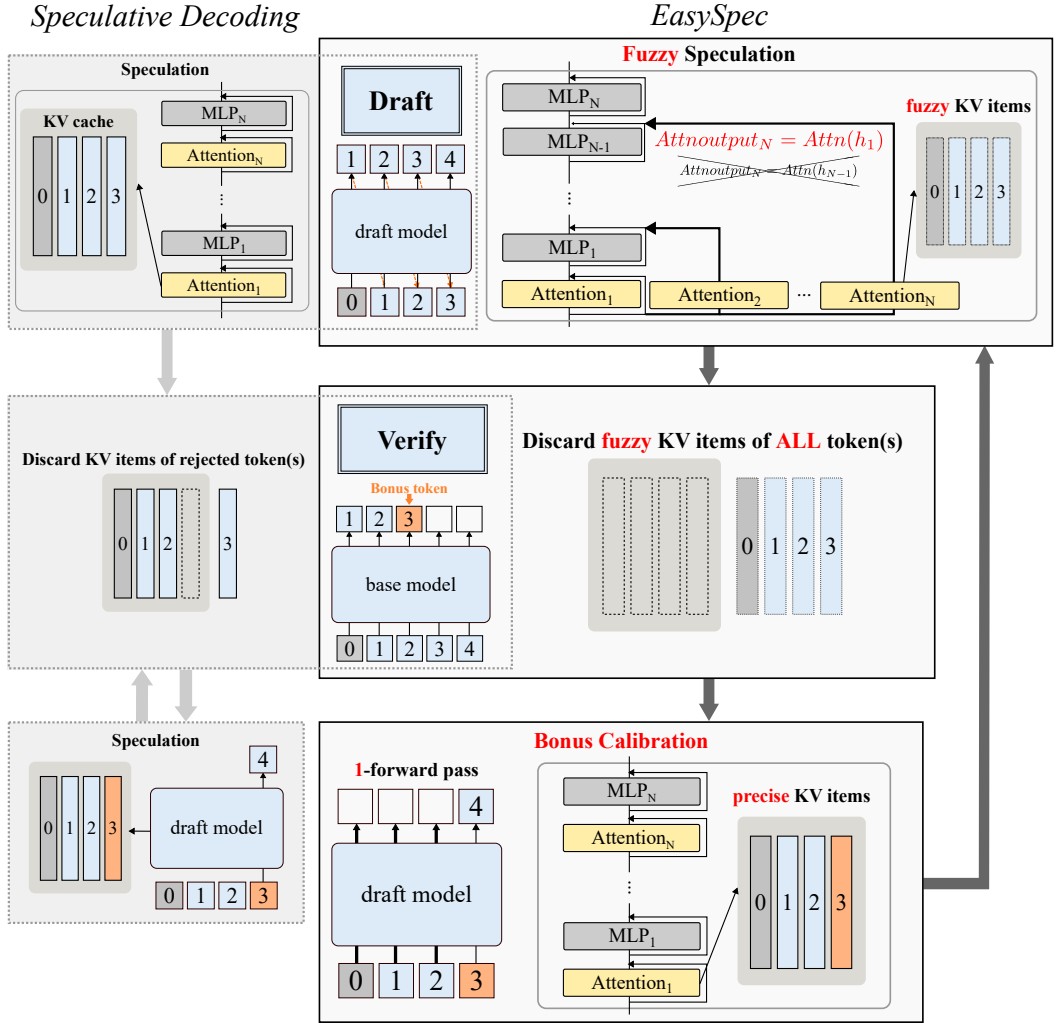

Figure 1: Overview of EasySpec: comparison between standard Speculative Decoding and EasySpec. The differences are mainly about: (1) Fuzzy Speculation uses layer parallelization for drafting acceleration. (2) The fuzzy KV items are discarded regardless of acceptance length, while Speculative Decoding preserves accepted items. (3) Speculative Decoding is 2-staged and EasySpec is 3-staged, with an additional stage for bonus calibration.

the draft model is typically smaller than that of the base model, the optimal TP size (the number of segments of workload distribution) is correspondingly smaller [12], meaning that the draft model would run fastest when dispatched on one or a subset of GPUs, leaving other GPUs idle (see Figure 2 and Table 6). Consequently, multi-GPU computational resources are under-utilized during the drafting stage.

We identify the primary cause of such inefficiency as the lack of parallelism between the draft model's layers: while tensor operations within one layer can be parallelized by TP, the layers themselves are restricted to be executed sequentially, one after another and from bottom to top, for generating a "precise" result of inference. However, the drafting result is never required to be precise, as it is only used for token parallelism and does not directly impact the final output—the verification result does (see Section 2.1). Therefore, strictly following the execution order is unnecessary, while a "fuzzy" but faster layer execution strategy could be preferable than the precise one, as long as it can sufficiently approximate the drafting result.

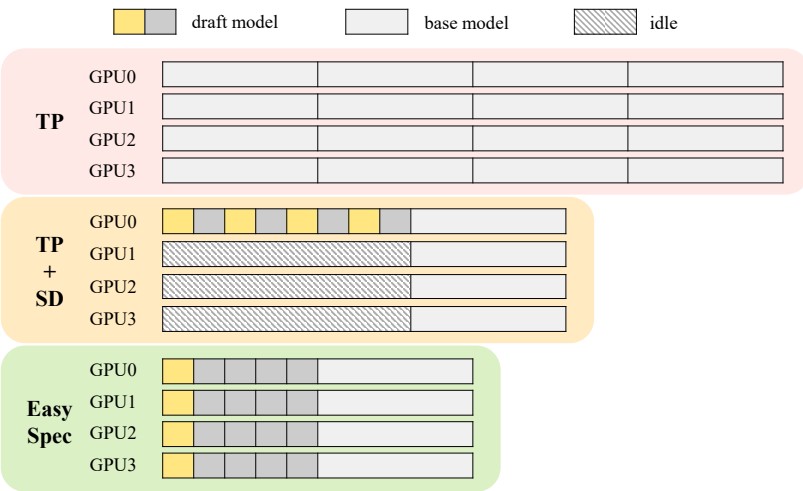

Figure 2: GPU runtime of different decoding algorithms in the distributed system. Speculative Decoding (SD) enables token-level parallelism (fewer base-model blocks), while causing multi-GPU under-utilization during the drafting stage. EasySpec solves the problem by layer-parallel speculation.

Based on this insight, we propose EasySpec, a layer-parallel speculation strategy for optimizing the efficiency of drafting-stage multi-GPU utilization. EasySpec introduces **fuzzy speculation**, which breaks the data dependencies between some consecutive attention layers by running them with the most recent hidden state as input to all of them (Figure 1 and Algorithm 2). As the data dependencies are eliminated, multiple layers of the draft model can execute simultaneously on separate devices, achieving layer-level parallelism, more efficient multi-GPU utilization and hence faster speculation inference. Meanwhile, the speculation outputs can be well-approximated, retaining a high speculation accuracy.

The approximation errors in the key-value (KV) cache of the draft model may accumulate during the inference procedure. To prevent this, we perform **bonus calibration** after each iteration of drafting-and-verification. Firstly, the accepted tokens are concatenated to the bonus token to form a token sequence. This sequence is then re-input to the draft model, where a conventional layer-sequential forward pass is executed, and the KV cache is updated with the precise values. This calibration step applies token parallelism, which is typically used only in the verification stage, to the drafting stage as well. Therefore, it incurs minimal additional latency to the speculation procedure, while significantly enhancing accuracy—ultimately contributing to improved overall inference speed.

We evaluate our method on several widely-used open-source LLMs and task-specific models. The draft models are selected from the same series as their corresponding large models. The evaluation results show that EasySpec can achieve a peak speedup of 4.17x over vanilla decoding. Specifically, the drafting stage can be accelerated by up to 1.62x with no greater than 7% drop of speculation accuracy, requiring no additional training or fine-tuning on the existing draft models.

## 2 Preliminary

### 2.1 Speculative decoding

Speculative decoding is a two-stage non-autoregressive decoding method for inference acceleration. The two stages are namely drafting and verification, which are iteratively executed as the inference proceeds. At time step $t$, the input token sequence is $X$. In the drafting stage, a smaller and faster model $M'$ is employed as the draft model. $M'$ auto-regressively runs $n$ times, generating a speculation token sequence $X, x'_{t+1}, ..., x'_{t+n}$ and probability distributions $p'_{t+1}, ..., p'_{t+n}$. The original large model $M$ then takes the whole sequence as input and conducts a single-forward verification, the outputs of which are $p_{t+1}, ..., p_{t+n+1}$. The acceptance probability of token $x'_i$ is $\min(1, \frac{p_i(x'_i)}{p'_i(x'_i)})$. After the verification sampling, the sequence of $m$ accepted tokens $x_{t+1}, ..., x_{t+m}$

(the same as $x'_{t+1}, ..., x'_{t+m}$) will be appended to the final output, along with a "bonus" token $x_{t+m+1}$ (called bonus because it is a by-product of the verification process) which can be sampled from the distribution $p$:

$$p = \begin{cases} norm(\max(0, p_{t+m+1} - p'_{t+m+1})) & (m < n) \\ p_{t+m+1} & (m = n) \end{cases} \tag{1}$$

Through this whole verification process, the final output token sequence aligns with the original distribution of the base model. Details of this algorithm are in Section A.

## 2.2 Tensor-parallel distributed inference

Tensor parallelism is a technique for deploying large models across multiple devices (GPUs). As for TP distributed inference, tensors are split up into multiple chunks, and each of these chunks is dispatched on a unique GPU. During computation operations, each GPU works on its respective chunk independently at first, and the results are synchronized afterwards to ensure that the final result is identical to the original. In modern multi-GPU systems, advanced hardware and communication protocols enable TP to be applied for inference acceleration.

The optimal TP size is primarily influenced by the parameter size. Firstly, the TP size must evenly divide both the hidden dimension and the number of attention heads. For example, in Qwen-2-0.5B-Instruct these numbers are 4864 and 14 respectively, which limits the maximum feasible TP size to 2. Secondly, the computational workload varies with parameter size, which further impacts TP efficiency. A detailed discussion of these device-level characteristics is provided in Section 2.3.

## 2.3 Device characteristics

Modern GPUs feature powerful computation capacities and hence short computation latency. Therefore, matrix computation involved in LLM inference is usually bottlenecked by memory bandwidth or control-flow overheads [14]. In multi-GPU systems, the computation can also be bounded by communication and synchronization overheads.

Formally, we denote the workload of 1-token forward as $W$, the latency of executing $W$ on a single GPU as $T_{exe}(W)$, and the additional overhead of TP distribution as $T_{addi}$. The workload of $s$-token parallelism is $sW$, and $W/s$ the workload of $s$-sized TP. If $W$ is relatively small compared to the computation power, we can expect the following relationships:

$$T_{exe}(W) \approx T_{exe}(sW) < sT_{exe}(W) \tag{2}$$

$$T_{exe}(W/s) + T_{addi} > T_{exe}(W) \tag{3}$$

The effectiveness of token-level parallelism can be illustrated by Equation (2), which is leveraged by speculative decoding for acceleration. Equation (3) highlights the impact of communication and control-flow overheads in distributed inference, which constrains the TP size not to be too large and limits multi-GPU utilization of the draft model.

## 3 Method

In this section, we present the details of how EasySpec enhances multi-GPU parallelism to accelerate the drafting stage of speculative decoding. The two key steps—fuzzy speculation and bonus calibration—are specified within each segment.

## 3.1 Fuzzy speculation

As mentioned in Section 1, the conventional speculation process restricts the layers to be executed sequentially. Algorithm 1 demonstrates the execution procedure of $N$ consecutive layers. If we eliminate $h'_i$ in the algorithm, we have

$$h_{i+1} = h_i + Attnoutput_i + MLPoutput_i \tag{4}$$

, which illustrates the data dependencies between the input of layer $i + 1$ and the output of layer $i$. In other words, before the outputs of the previous attention layer ($Attnoutput_i$) and MLP layer

---
**Algorithm 1** Layer-Sequential Speculation

  **Input:** hidden state $h$, $N$ consecutive attention layers $Attn_1,\cdots,Attn_N$ and MLP layers $MLP_1,\cdots,MLP_N$
  $h_1 = h$
  **for** $i = 1$ **to** $N$ **do**
    $Attnoutput_i = Attn_i(h_i)$ (sequential)
    $h'_i = h_i + Attnoutput_i$
    $MLPoutput_i = MLP_i(h'_i)$
    $h_{i+1} = h'_i + MLPoutput_i$
  **end for**

---

---
**Algorithm 2** Layer-Parallel Fuzzy Speculation

  **Input:** hidden state $h$, $N$ consecutive attention layers $Attn_1,\cdots,Attn_N$ and MLP layers $MLP_1,\cdots,MLP_N$
  $h_1 = h$
  **for** $i = 1$ **to** $N$ **do**
    $Attnoutput_i = Attn_i(h_1)$ (parallel)
  **end for**
  **for** $i = 1$ **to** $N$ **do**
    $h'_i = h_i + Attnoutput_i$
    $MLPoutput_i = MLP_i(h'_i)$
    $h_{i+1} = h'_i + MLPoutput_i$
  **end for**

---

($MLPoutput_i$) are computed, nothing can be done with the upper layer $i + 1$ (and $i + 2$ to $N$ as well).

Such a restriction on the execution order is necessary when the draft model is directly used for inference. Executing the model differently would shift the output probability distributions $p'$ to $p''$, undermining the quality of generated content. However, the situation is different when the draft model is just used for speculation. Regardless of how $p''$ deviate from the original, as long as we set the acceptance probabilities to $\min(1, p/p'')$, the final output distribution will theoretically remain the same as the base model. This leads to a potential optimization of speculation strategy: if the output distribution $p'$ can be efficiently approximated by $p''$, such that $p'' \approx p'$, the overall process may speedup, while the final outputs remain lossless. That is to say, a slightly fuzzy but faster approach for speculation could outperform the precise one.

We propose an effective approximation method for fuzzy speculation. Inspired by [15], we observe that the attention layers' outputs can be well approximated by substituting the input $h_i$ with $h_1$ for all $1 < i \leq N$. In Equation (4), the hidden state update $h_{i+1} - h_i = Attnoutput_i + MLPoutput_i$ is relatively less significant, so the cosine similarity between $h_i$ and $h_{i+1}$ can approach 1, and consequently the similarity between all $h_i$ and $h_1$ is also high (Figure 3 is a 2-D demonstration of this phenomenon). Such high similarities of the input allow other hidden states to be accurately approximated as well (see Section E.1 for theoretical explanations and Table 4 for empirical results).

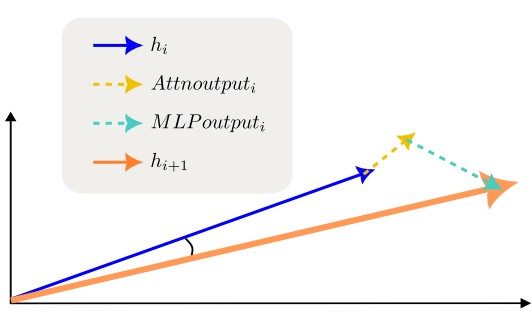

Figure 3: 2-D demonstration of high cosine similarity between $h_i$ and $h_{i+1}$.

Algorithm 2 shows the detailed steps of this method. From the perspective of model architectures, the input of the attention layers is not the most recent hidden states, but rather those from several layers prior (Figure 1). With this modification of layer execution strategy, we break the data dependencies between $N$ consecutive attention layers: once $h_j$ is computed for some $j$, all attention layers between $j$ to $j + N$ can be executed with no data dependencies among them, thereby achieving a layer-level concurrency. In a multi-GPU system, these attention layers can run simultaneously on different devices (Figure 2).

Denote the execution time of sequential and fuzzy speculation over these $N$ attention layers as $T_{seq}$ and $T_{fuzzy}$, we have

$$T_{seq} = NT_{exe}(A)$$
$$T_{fuzzy} = T_{exe}(A) + T_{addi} \tag{5}$$

where $A$ is the workload of one attention layer. Unlike the traditional TP method, the latency of running an attention layer is long enough to compensate for the additional overhead, that is

Table 1: Example layer-parallel strategies for drafter models. The strategies for other drafters is the same as specified in Section 3.1.1.

| Model | # of Layers | $N$ | Strategy |
|---|---|---|---|
| Llama-3-8B-Instruct | 32 | 4 | 0, 1-3, 4-7, 8-11, ..., 24-27, 28-30, 31 |
| Llama-3.2-1B-Instruct | 16 | 2 | 0, 1-2, 3-4, 5-6, ..., 11-12, 13-14, 15 |
| Qwen2-7B-Instruct | 28 | 4 | 0, 1-3, 4-7, 8-11, ..., 20-23, 24-26, 27 |

$(N-1)T_{exe}(A) >> T_{addi}$ for $N > 1$, resulting in $T_{fuzzy} < T_{seq}$. The MLP layers are executed in the original sequential order, as parallelizing them results in significant approximation errors and degraded end-to-end performance (Section B).

In order to achieve a higher acceptance rate, EasySpec also incorporates tree attention [16] in the drafting stage, a technique commonly used in existing speculative decoding methods [7, 17, 18]. Tree attention can boost the acceptance rate by increasing the number of draft token sequences. As shown in Figure 5, the drafted tokens are structured as a tree, with each path corresponding to a sequence. Speculation and verification for all the sequences can be performed in a single forward pass using a 2D tree-attention mask.

### 3.1.1 Layer-parallel strategy

For the layer-parallel size $N$, we exclude the first and last layer and parallelize $1 \sim N - 1$, $N \sim 2N - 1$, ..., until the remaining layers (except for the last) are not enough to compose a $N$-set. For instance, for Llama-3-8B-Instruct (32 layers) and $N = 4$, we have $1 \sim 3$, $4 \sim 7$, ..., $28 \sim 30$ parallelized, while layer 28,29, and 30 have to compose a 3-set because layer 31 is the last (see Table 1). The reason of such strategy is that, parallelizing intermediate layers (excluding the first and last layer) yields best performances in our experiments. We hypothesize that this is because the first and last layer are more sensitive to approximation errors.

Note that there could be a better strategy for models and/or tasks, but this strategy has already yielded sufficiently satisfactory acceleration. Moreover, trying other strategies is simple and lightweight, as it requires no modifications to the original model architecture. A change in the configuration is enough.

### 3.2 Bonus calibration

Formally, the speculation of token $j$ can be denoted as $x'_j, p'_j, kv'_{j-1} = M'(x'_{i<j-1}, kv'_{i<j-1}, x'_{j-1})$. Fuzzy speculation induces errors to the value of $x'$, $p'$ and $kv'$. The errors in $x'$ and $p'$ are limited within a single draft-verify iteration, since the rejected tokens and all output distributions will be discarded after verification. However, the errors in $kv'_{j-1}$ will affect all subsequent generations after $j$, leading to long-term error accumulation that can severely degrade the accuracy.

The precise values of $kv'$ are required for preventing such error accumulation. Nevertheless, obtaining these values demands running the model's layers sequentially, which is exactly what we try to optimize through fuzzy speculation. To solve this dilemma, we propose an efficient way of obtaining precise KV values with minimal additional latency. At the beginning of a standard drafting stage, the bonus token $x_{t+m+1}$ from the last iteration is input to the draft model, for the speculation of $x'_{t+m+2}$. This is a 1-token forward pass and also memory-bounded, as discussed in Section 2.1. Therefore, similar to the verification stage, token parallelism can also be applied here. By concatenating the accepted token sequence $x_{t+1}, \cdots, x_{t+m}$ with the bonus token $x_{t+m+1}$ and re-inputting this entire sequence $x_{t+1}, \cdots, x_{t+m}, x_{t+m+1}$ to the draft model, the precise KV values $kv'_{t+1}, \cdots, kv'_{t+m}$ (and $kv'_{t+m+1}$) can be produced through a single sequential forward pass. According to Equation (2), the latency of such a forward is nearly equal to a 1-token sequential forward.

We propose bonus calibration based on the method above. As shown in Figure 1, we firstly discard all fuzzy KV items, regardless of whether the corresponding tokens are accepted or rejected (e.g. the accepted items 1 and 2 are also discarded). Then, a token-parallel forward is conducted to refill the KV cache with precise values. From the perspective of execution, the precise KV values are by-products of the next-token speculation of the bonus token, similar to how the bonus token itself is produced by the verification. This is the reason why we call this process "bonus". Bonus calibration can generate a precise candidate token and distribution, while subsequent rounds of speculation can

revert to be fuzzy. The overall latency increases slightly, from $nT_{fuzzy}$ to $T_{seq} + (n-1)T_{fuzzy}$ ($n$ is the speculation length), while the elimination of errors is crucial and substantial. More detailed theoretical backing is in Section E.2.

# 4 Experiment

**Models and benchmarks.** We evaluated EasySpec on Llama-3-70B-Instruct and Qwen2-72B-Instruct, as well as Llama-3.3-70B-Instruct and task-specific models Qwen2-Math-72B-Instruct and Qwen2.5-Coder-32B-Instruct [19]. The 0.5B/1.5B/3B/7B/8B models in the same series serve as the draft models for their respective base models. The benchmarks include a variety of tasks: language understanding (MMLU[20]), code generation (HumanEval[21]), math reasoning (MATH[22]), instruction following (IFEval[23]) and multilingual language usage (MGSM[24]). All the experiments were conducted using chain-of-thought reasoning with a maximum of 128 tokens. We also evaluated our method on Spec-Bench [25] for comparison with related work.

**Environments and configurations.** The experiments were conducted on 8×A100 GPUs. Unless specified otherwise, the layer-parallel size $N$ is set to 4, as it is optimal for most of the tested models. The tensor-parallel sizes for drafter and base models are 1 and 8, as they are the optimal in most cases (see Table 6). The optimal and applied speculation length is 5.

**Baseline settings.** We use the following baselines for comparison: tensor-parallel distributed decoding (TP), speculative decoding in the distributed system (+SD), and tree attention (+tree) [16]. Additionally, we compare our work with EAGLE-2 [18].

**Metrics.** We use token throughput of single-batch inference as the performance metric. We also measure the running time per 100 tokens of the drafting and verification stages, which demonstrates the effectiveness of acceleration more clearly. Acceptance rates are also recorded and presented.

## 4.1 Main results

### 4.1.1 Effectiveness

To illustrate the robust effectiveness of EasySpec, we conducted experiments on Llama-3-70B-Instruct and Qwen2-72B-Instruct across datasets, with temperatures=0 and 0.8. The results are shown in Table 2. We measure the running time per 100 tokens of the drafting (d) and verification (v) stages and the total acceleration ratio (total) compared to vanilla decoding. $\alpha$ represents the acceptance rates. The exact total running time of all baselines are in Table 9.

The results show that EasySpec consistently accelerates the overall execution of speculative decoding across all datasets. While the verification stage (the large models) can be accelerated by TP, the drafting stage often becomes the primary bottleneck, accounting for a large portion of the total running time. EasySpec accelerates the drafting stage by up to 1.62x, thereby improving the overall throughput. Meanwhile, the drop of speculation accuracy is no more than 7%, suggesting that fuzzy speculation with bonus calibration is sufficiently accurate across all evaluated cases.

We also evaluated EasySpec on two task-specific models: Qwen2-Math-72B-Instruct and Qwen2.5-Coder-32B-Instruct, along with some other combinations of drafting and base models. The effectiveness of EasySpec in these cases are shown in Table 7 and Table 8.

### 4.1.2 Generalization and stability

An intermediate question is that whether we can use an even shallower model to reduce drafting latency. Firstly, such a model often does not exist within the same series, while training a draft model from scratch is non-trivial and the trained drafter often lacks generalization capability across base models and datasets [26]. Secondly, an overly shallower drafter tends to exhibit unstable speculation performance, while smaller versions of model in the same series inherently have alignments in behaviors with the larger, due to their shared tokenizer, pretraining corpora and similar training process [25].

To clarify the above statements, we compare our method with EAGLE-2 [18], a typical tiny-drafter speculative decoding method. EAGLE-2 uses a well-trained one-layer draft model, aiming for extreme drafting latency reduction. We evaluated both methods on Spec-Bench [25]. Besides average token

Table 2: Running time per 100 tokens, total acceleration ratio and acceptance rates of two mainstream models across datasets. 'd', 'v' stands for drafting and verification running time (s/100 tokens) respectively, and 'total' and $\alpha$ stands for total acceleration ratio and acceptance rates. More detailed numbers are in Table 9.

| Dataset | Method | Llama-3-70B(8B)-Instruct | | | | Qwen2-72B(7B)-Instruct | | | |
| | | d | v | total | $\alpha$ | d | v | total | $\alpha$ |
|---|---|---|---|---|---|---|---|---|---|
| | | | | | temperature=0 | | | | |
| MMLU | TP | - | - | 1.53x | - | - | - | 1.56x | - |
| | +sd | 3.52 | 2.15 | 2.05x | 0.57 | 3.32 | 2.25 | 2.13x | 0.52 |
| | +tree | 2.52 | 1.59 | 2.82x | 0.88 | 2.38 | 1.62 | 2.96x | 0.85 |
| | EasySpec | 1.70(↑1.48x) | 1.73 | **3.38x** | 0.82 | 1.65(↑1.44x) | 1.70 | **3.55x** | 0.80 |
| HE | TP | - | - | 1.55x | - | - | - | 1.57x | - |
| | +sd | 2.93 | 1.79 | 2.50x | 0.74 | 2.82 | 1.83 | 2.58x | 0.69 |
| | +tree | 2.53 | 1.58 | 2.87x | 0.92 | 2.26 | 1.51 | 3.18x | 0.95 |
| | EasySpec | 1.61(↑1.57x) | 1.63 | **3.64x** | 0.87 | 1.48(↑1.52x) | 1.54 | **3.97x** | 0.91 |
| MATH | TP | - | - | 1.52x | - | - | - | 1.54x | - |
| | +sd | 2.96 | 1.74 | 2.45x | 0.73 | 2.48 | 1.65 | 2.86x | 0.78 |
| | +tree | 2.50 | 1.47 | 2.90x | 0.95 | 2.20 | 1.45 | 3.24x | 0.96 |
| | EasySpec | 1.58(↑1.58x) | 1.55 | **3.68x** | 0.91 | 1.44(↑1.52x) | 1.47 | **4.06x** | 0.95 |
| IFEval | TP | - | - | 1.50x | - | - | - | 1.52x | - |
| | +sd | 3.68 | 2.16 | 1.93x | 0.55 | 4.24 | 2.80 | 1.64x | 0.39 |
| | +tree | 2.53 | 1.55 | 2.76x | 0.89 | 2.80 | 1.84 | 2.49x | 0.72 |
| | EasySpec | 1.68(↑1.51x) | 1.65 | **3.39x** | 0.82 | 1.87(↑1.50x) | 1.94 | **3.04x** | 0.67 |
| MGSM | TP | - | - | 1.54x | - | - | - | 1.56x | - |
| | +sd | 2.66 | 1.61 | 2.73x | 0.80 | 2.62 | 1.75 | 2.73x | 0.72 |
| | +tree | 2.45 | 1.48 | 2.96x | 0.96 | 2.12 | 1.50 | 3.29x | 0.94 |
| | EasySpec | 1.55(↑1.58x) | 1.51 | **3.80x** | 0.93 | 1.54(↑1.37x) | 1.57 | **3.83x** | 0.88 |
| | | | | | temperature=0.8 | | | | |
| MMLU | TP | - | - | 1.53x | - | - | - | 1.56x | - |
| | +sd | 3.09 | 1.93 | 2.32x | 0.67 | 2.90 | 1.93 | 2.47x | 0.65 |
| | +tree | 2.49 | 1.54 | 2.89x | 0.94 | 2.14 | 1.48 | 3.30x | 0.95 |
| | EasySpec | 1.59(↑1.56x) | 1.63 | **3.61x** | 0.89 | 1.49(↑1.43x) | 1.54 | **3.94x** | 0.93 |
| HE | TP | - | - | 1.54x | - | - | - | 1.59x | - |
| | +sd | 2.94 | 1.81 | 2.48x | 0.73 | 2.54 | 1.63 | 2.91x | 0.80 |
| | +tree | 2.47 | 1.49 | 2.98x | 0.96 | 2.08 | 1.45 | 3.44x | 0.97 |
| | EasySpec | 1.53(↑1.62x) | 1.49 | **3.90x** | 0.94 | 1.45(↑1.44x) | 1.46 | **4.17x** | 0.97 |
| MATH | TP | - | - | 1.53x | - | - | - | 1.55x | - |
| | +sd | 2.81 | 1.65 | 2.59x | 0.78 | 2.44 | 1.54 | 2.98x | 0.85 |
| | +tree | 2.44 | 1.45 | 2.97x | 0.97 | 2.09 | 1.40 | 3.39x | 0.99 |
| | EasySpec | 1.52(↑1.61x) | 1.48 | **3.86x** | 0.94 | 1.43(↑1.47x) | 1.44 | **4.13x** | 0.98 |
| IFEval | TP | - | - | 1.50x | - | - | - | 1.51x | - |
| | +sd | 3.08 | 1.87 | 2.29x | 0.67 | 3.24 | 2.14 | 2.16x | 0.56 |
| | +tree | 2.45 | 1.46 | 2.89x | 0.97 | 2.17 | 1.51 | 3.16x | 0.91 |
| | EasySpec | 1.52(↑1.61x) | 1.50 | **3.73x** | 0.92 | 1.53(↑1.42x) | 1.58 | **3.73x** | 0.87 |
| MGSM | TP | - | - | 1.55x | - | - | - | 1.57x | - |
| | +sd | 2.54 | 1.55 | 2.85x | 0.84 | 2.68 | 1.72 | 2.73x | 0.76 |
| | +tree | 2.30 | 1.44 | 3.13x | 0.98 | 2.08 | 1.45 | 3.41x | 0.97 |
| | EasySpec | 1.52(↑1.51x) | 1.49 | **3.88x** | 0.96 | 1.46(↑1.42x) | 1.50 | **4.06x** | 0.94 |

throughput, we also calculate the variances of throughput across data items, for better demonstration of performance stability. The temperature is set to 0.8 for both methods.

The results in Table 3 show that, for Llama-3-70B-Instruct, the EasySpec-accelerated 1B drafter achieves higher average token throughput than EAGLE-2 (≈1B) across all tasks, suggesting that a smaller model from the same series, if existing, is good enough to deliver satisfactory performance with EasySpec. As for performance stability, EasySpec exhibits significantly smaller variances, which should be attributed to its remarkably higher speculation accuracies (nearly 0.8 vs less than 0.4). For Llama-3.3-70B-Instruct, the performance gap becomes even more evident, further indicating that trained drafters lack generalization capability to other base models, whereas smaller models from the

Table 3: Average token throughput(variance) and acceptance rates of EasySpec and EAGLE-2 for Llama-3-70B-Instruct on Spec-Bench. The format of throughput is mean(variance). $\alpha$ stands for acceptance rates.

| | EAGLE-2 | | EasySpec | | EAGLE-2 | | EasySpec | |
|------|---------|---|----------|---|---------|---|----------|---|
| | Llama-3-70B-Instruct | | | | Llama-3.3-70B-Instruct | | | |
| Task | Throughput | $\alpha$ | Throughput | $\alpha$ | Throughput | $\alpha$ | Throughput | $\alpha$ |
| MT | 37.15(31.27) | 0.38 | 39.81(10.95) | 0.85 | 25.92(14.28) | 0.24 | 38.69(23.1) | 0.84 |
| TL | 32.97(17.28) | 0.38 | 35.31(15.41) | 0.72 | 19.05(8.46) | 0.15 | 34.14(16.06) | 0.71 |
| SUM | 36.34(9.30) | 0.40 | 39.18(4.88) | 0.80 | 24.23(3.63) | 0.20 | 37.67(5.03) | 0.77 |
| QA | 31.85(20.15) | 0.34 | 40.38(9.82) | 0.84 | 22.50(10.36) | 0.19 | 40.30(3.41) | 0.84 |
| MR | 38.10(14.97) | 0.45 | 43.52(4.44) | 0.93 | 27.09(6.14) | 0.25 | 44.59(2.49) | 0.96 |
| RAG | 37.26(38.15) | 0.47 | 38.15(12.56) | 0.83 | 23.25(15.40) | 0.22 | 36.79(13.75) | 0.80 |

Table 4: Average cosine similarities between precise and approximated hidden states. The tested model and task are Llama-3-8B-Instruct and MMLU.

| LP size | $h$ | $q$ | $k$ | $v$ | $Attnoutput$ |
|---------|------|------|------|------|--------------|
| 2 | 0.93 | 0.98 | 0.99 | 0.92 | 0.93 |
| 3 | 0.89 | 0.97 | 0.98 | 0.86 | 0.88 |
| 4 | 0.86 | 0.96 | 0.97 | 0.82 | 0.83 |

same series remain compatible. Note that EasySpec requires no training or fine-tuning on either the target distribution or fuzzy approximation, making it a plug-in method of higher adaptability.

## 4.2 Approximation precision

The feasibility of fuzzy speculation relies on the approximation precision of the draft model, which is closely related to the cosine similarities between the precise and fuzzy-approximated hidden states. Table 4 shows average cosine similarities between these hidden states. The tested drafter is Llama-3-8B-Instruct and the benchmark is MMLU. The evaluated hidden states include input hidden states $h$, outputs of attention layers $Attnoutput$, and queries($q$), keys($k$) and values($v$) inside attention heads. We set layer-parallel sizes to 2,3,4 for diverse approximation granularity. The results show that, as more layers are parallelized, the cosine similarity decreases but stays above 0.8, indicating a sufficiently high precision of approximation. Notably, the queries and keys are well-approximated with cosine similarities approaching 1. This is critical for maintaining precision of other hidden states and outputs, as errors in attention weights scale exponentially.

## 4.3 Ablation study

We conducted an ablation study on 3 aspects of EasySpec: tree attention width, layer-parallel (LP) size and the presence or absence of bonus calibration. We varied the tree width across 1,4,8,12 and LP size from 1 to 5. The results of these different configurations are presented in Figure 4.

For token throughput (left part), as the LP size increases from 1, the throughput improves due to better multi-GPU utilization of fuzzy speculation. However, when the LP size becomes too large (e.g. 5), the speculation results are excessively fuzzy, leading to a noticeable decline in token throughput. The optimal LP size is influenced by the tree width: a wider tree increases the likelihood of existence of correct tokens, thus allowing for a fuzzier and faster speculation (a larger LP size). Moreover, even without tree attention (width=1), EasySpec can also accelerate the inference with layer parallelism (see the green line in Figure 4).

For speculation accuracy, the right part shows that bonus calibration can significantly improve acceptance rates for every combination of tree width and LP size. Bonus calibration also slows the decline in acceptance rates with increasing LP sizes (see the right part of Figure 4). All the above

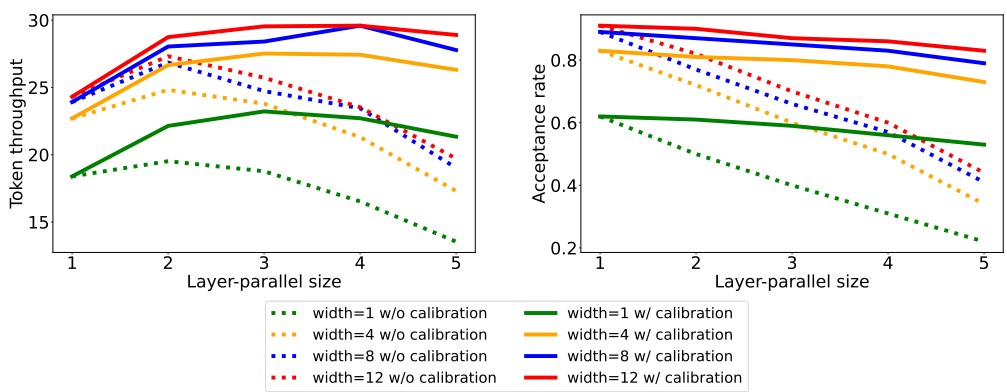

Figure 4: Token throughput and acceptance rates of Llama-3-70B(8B)-Instruct on MMLU under different configurations of EasySpec. Left: token throughput. Right: acceptance rates.

results prove that the trade-off between precision of KV values and additional latency is reasonable and effective.

## 5 Related work

Methods focusing on the drafter stage can be broadly categorized into training drafters, self-speculation or algorithmic optimization. Specinfer [16] uses boost-trained draft models and tree-structured attention for efficient speculation. Medusa [7] trains a set of extra MLP heads for future token prediction using the original LLM's features. Self-Speculative Decoding [27] and LayerSkip [28] skips some layers of the target LLM for self-speculation. Lookahead [26] uses n-gram Jacobi decoding to increase acceptance rates. REST [29] leverages a data storage for retrieval speculation. Recent work also explores optimized tree-structured attention mechanisms. SpecExec [30] takes the most probable continuations from the draft model to build a cache tree, improving inference efficiency on resource-constrained GPUs. OPT-Tree [31] proposes an adaptive and scalable draft-tree construction algorithm that maximizes the expected acceptance length. Beyond the drafter itself, other studies investigate stage-level parallelism and theoretical explanations of speculative decoding. For instance, PEARL [32] overlaps the drafting and verification stages to achieve stage-level pipeline, while SpecTr [33] adopts optimal transport theory to provide a theoretical explanation and optimization.

The discrepancy between optimal distribution sizes is discovered by the authors of [12], who suggested training a wider but shallower draft model for better performance under tensor parallelism. There are currently few works focusing on this problem, yet it is important and worth-solving. To the best of our knowledge, we are the first to effectively optimize multi-GPU utilization of currently-available draft models.

## 6 Conclusion

In this paper, we propose EasySpec, a layer-parallel speculation strategy for efficienct multi-GPU utilization during the drafting stage of speculative decoding. EasySpec introduces two modifications to the original speculation process: fuzzy speculation and bonus calibration. Fuzzy speculation breaks the sequential layer execution order and enables multi-layer parallelization, while bonus calibration applies token parallelism to the drafting stage to eliminate long-term error accumulation. EasySpec is training-free and consistently achieves acceleration across models and datasets.

## Acknowledgments

This work is partially supported by the NSF of China (under Grant 92364202), and Major Program of ISCAS (Grant No. ISCAS-ZD-202402).

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

# A Speculative decoding

Speculative decoding consists of iterations of drafting and verification. The detailed algorithm of a single iteration is present in Algorithm 3.

---

**Algorithm 3** An Iteration of Speculative Decoding

---

**Input:** input token sequence $X$, drafting length $m$, draft model $M'$, base model $M$
**for** $i = 1$ **to** $m$ **do**
   $x_i', p_i' = M'(X, x_{j<i}')$
**end for**
$p_1, \cdots, p_m, p_{m+1} = M(X, [x_1', \cdots, x_m'])$
sample $r_1, \cdots, r_m$ independently from $U(0, 1)$
**for** $i = 1$ **to** $m$ **do**
  **if** $r_i > \frac{p_i(x_i')}{p_i'(x_i')}$ **then**
    $n = i - 1$ ($n$ is the accepted length)
    **break** (rejected)
  **else**
    $x_i = x_i'$ (accepted)
  **end if**
**end for**
**if** $n < m$ **then**
  $p_{bonus} = norm(\max(0, p_n - p_n'))$
**else**
  $p_{bonus} = p_{n+1}$
**end if**
sample $x_{bonus}$ from $p_{bonus}$
**return** $X, x_1, \cdots, x_n, x_{bonus}$

---

For any distribution $p'$, the final output token sequence is equivalent to sampling them directly from $p$. The proof is specified in many existing works (e.g. [11]).

# B Strategy for MLP layers

In EasySpec, the MLP layers are executed sequentially, rather than parallelized like the attention layers. This design choice stems from the empirical results that parallelizing all layers (both attention and MLP) introduces substantial approximation errors, leading to degraded overall performance. As shown in Table 5, even parallelizing just $N = 2$ full layers results in lower token throughput and reduced speculation accuracies compared to the optimal attention-only parallelization strategy, with performance deteriorating further as $N$ increases.

# C Tree attention

In the context of speculative decoding, tree attention can be leveraged to speculation and verification for multiple possible token sequences in a single forward pass. With a 2-dimensional attention mask

Table 5: Token throughput and speculation accuracies of models with different parallel strategies on MMLU.

| Model | Parallelized Layer | $N$ | Token throughput | $\alpha$ |
|---|---|---|---|---|
| Llama-3-70B(1B)-Instruct | Only Attention | 2 | **34.37** | **0.77** |
| | Full Layer | 2 | 29.71 | 0.58 |
| Qwen2-72B(1.5B)-Instruct | Only Attention | 4 | **29.94** | **0.76** |
| | Full Layer | 2 | 27.37 | 0.68 |
| | Full Layer | 3 | 24.10 | 0.51 |

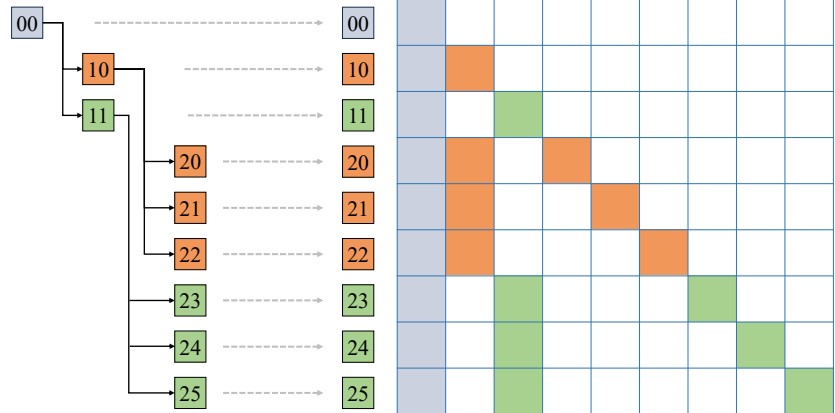

Figure 5: Illustration of tree attention. 6 draft token sequences are generated within 2 forward passes, which increases parallelism and potentially acceptance rates. The whole tree can be verified in a single-forward pass of the base model.

Table 6: Token throughput of models at different TP sizes on MMLU. '*' means the results are from vLLM.

| Model | TP=1 | TP=2 | TP=4 | TP=8 |
|---|---|---|---|---|
| L3-70B | 8.39 | 13.00 | 13.23 | 13.23 |
| L3-70B* | OOM | 19.5 | 28.47 | 28.25 |
| Q2-72B | 8.49 | 13.03 | 13.01 | 12.89 |
| Q2-72B* | OOM | 18.91 | 28.63 | 29.39 |
| L3-8B | 36.76 | 33.90 | 32.31 | 32.39 |
| L3-8B* | 79.39 | 68.86 | 68.35 | 67.76 |
| Q2-7B | 37.16 | 36.46 | 37.12 | - |
| Q2-7B* | 83.38 | 73.9 | 73.81 | - |

applied to the flattened token tree, token sequences with the same prefix can share the KV-items, thus increasing token-level parallelism. As an illustration in Figure 5, with 2 forward passes, 6 draft token sequences are generated.

## D  Additional results

### D.1  Results of different TP sizes

Table 6 shows the token throughput of some typical tested models at different TP sizes. The benchmark consists of 40 data items from MMLU. While the large models benefit from multi-GPU distribution, the draft models experience slowdown when the TP size is greater than 1 (due to Equation (3)), confirming that the under-utilization of multi-GPU resources cannot be solved by traditional TP. We also tested the throughput in another widely-used inference engine vLLM [34], and the results are consistent with the above claim. This consistency supports the conclusion that the acceleration and slowdown are indeed attributable to multi-GPU configurations, rather than other possible factors. Based on these findings, we use 1 and 8 as the test-time distribution size for all small and large models respectively, without compromising generalization.

Note that Qwen-2-7B-Instruct does not support TP size=8, as its internal dimension size is not divisible by 8. That is also an aspect of lacking parallelism (other than the slowdown of inference speed) and can be solved by EasySpec as well.

Table 7: Running time per 100 tokens and acceptance rates of two task-specific models across datasets. The denotation is consistent with Table 2.

| Datasets | Methods | Q2.5-Coder-32B(7B)-Instruct | | | |
|---|---|---|---|---|---|
| | | d | v | total | $\alpha$ |
| Human Eval | vanilla | - | - | 7.13 | - |
| | +tree | 2.15 | 1.23 | 3.38 | 0.97 |
| | EasySpec | 1.43 | 1.23 | **2.66** | 0.97 |
| | | -0.72 | +0.00 | -0.72 | -0.00 |
| | | Q2-Math-72B(7B)-Instruct | | | |
| MATH | vanilla | - | - | 11.55 | - |
| | +tree | 2.05 | 1.51 | 3.55 | 0.99 |
| | EasySpec | 1.40 | 1.52 | **2.91** | 0.98 |
| | | -0.65 | +0.01 | -0.64 | -0.01 |

Table 8: Token throughput of other model combinations w/ and w/o EasySpec. The temperature is 0. The left and right numbers are throughput w/o and w/ EasySpec respectively. 'w/o EasySpec' is the same baseline as '+tree' in Table 2.

| Models | MMLU | | HumenEval | | MATH | | IFEval | | MGSM | |
|---|---|---|---|---|---|---|---|---|---|---|
| Q2-72B-1.5B | 24.02 | 29.94 | 26.28 | 33.78 | 27.54 | 36.11 | 20.65 | 25.17 | 25.92 | 33.26 |
| Q2-72B-0.5B | 24.57 | 29.29 | 28.80 | 33.15 | 29.61 | 35.82 | 20.57 | 24.08 | 27.48 | 32.63 |
| L3-70B-3B | 25.41 | 31.32 | 26.81 | 32.52 | 27.49 | 35.78 | 25.20 | 30.29 | 27.23 | 35.18 |
| L3-70B-1B | 32.10 | 34.37 | 34.25 | 37.25 | 35.70 | 40.00 | 28.60 | 34.04 | 35.78 | 40.25 |
| L3-8B-1B | 47.97 | 56.49 | 50.99 | 61.88 | 54.04 | 64.97 | 49.54 | 56.01 | 52.68 | 63.68 |

## D.2 Results of task-specific models

Table 7 shows the results of two task-specific models on the corresponding datasets MATH and HumanEval. On these datasets, the original speculation results of the draft models is already highly accurate (with the acceptance rates greater than 97%), indicating that the speculation is inherently simple. As a result, fuzzy speculation incurs virtually no drop in accuracies, achieving an ideal drafting-stage acceleration.

## D.3 Results of other combinations of draft and base models

We further tested EasySpec on other combinations of draft and base models. The results in Table 8 illustrates that EasySpec shows acceleration across all model combinations and datasets, highlighting the wide adaptability of our method. By 'without EasySpec', we mean that the baseline is '+tree' in Table 2. 'L3' and 'Q2' stand for Llama-3 and Qwen2 respectively. For the optimal overall performance, the layer-parallel size of Llama-3.2-1B-Instruct is 2, and the size of others is 4.

As stated in Section 4.1.2, using smaller drafters does not necessarily result in overall optimal performance. Furthermore, it should be noted that Qwen-2-1.5B and Qwen-2-0.5B only support TP size $\leq 4$ and 2, since the dimension size is not divisible by a larger number (similar to the Qwen 7B model), while EasySpec increases the parallel sizes by introducing extra layer-parallelism.

## D.4 Detailed numbers of results

Table 9 shows detailed numbers of results of EasySpec and baselines. Compared to Table 2, the total running time per 100 tokens, the running time of vanilla decoding are listed.

# E   Theoretical backing

## E.1   Connection between cosine similarity and approximation precision

Current transformer-based models typically apply a layer normalization (LayerNorm) operation before the attention block. Given two input hidden states, $h_1$ and $h_2$ (where $h_2$ is an approximation of $h_1$), the LayerNorm operator transforms them into normalized representations $h_1'$ and $h_2'$, which are then used in subsequent attention computations. Importantly, the LayerNorm operation preserves cosine similarity, i.e., $\cos(h_1, h_2) = \cos(h_1', h_2')$. When this cosine similarity is close to 1, the corresponding vectors are also close in Euclidean distance ($|h_1 - h_2| \approx 0$), ensuring that downstream computations maintain high numerical precision. For models without LayerNorm operation, the demonstration in Figure 3 also implies that the approximated hidden states can remain precise (in principle).

## E.2   Balance between speculation latency and accuracy

The strategy of speculative decoding (SD) is about finding an optimal balance between the running time of drafters $T_{draft}$ and the speculation accuracy $\alpha$. Denote the running time of base model as $T_{base}$, the overall running time of decoding $N$ step without speculative decoding is $N * T_{base}$. The overall running time with SD can be seen as a function of $T_{draft}$ and $\alpha$: $T_{all}(T_{draft}, \alpha) = N * T_{draft} + (N/(n * \alpha)) * T_{base}$ ($n$ is the speculation length), since the base model only needs to run $N/(n * \alpha)$ times. The key to our method is reducing $T_{draft}$ largely while maintaining a high $\alpha$, which will lead to decrease in the overall time.

Denote the running time of fuzzy speculation as $T_{fuzz}$ and original sequential run as $T_{ori}$. If we just run fuzzy speculation for every $n$ token, the time will be $T_{draft}' = n * T_{fuzz}$. Bonus calibration runs at every beginning of the speculation. As token parallelism is applied (just like it being applied to verification in normal SD), it has basically the same running time as $T_{ori}$. The full speculation time then becomes $T_{draft} = (n - 1) * T_{fuzz} + T_{ori} > n * T_{fuzz} = T_{draft}'$. If that were all the story, $T_{all}$ would increase. However, the $\alpha$ is largely increased by the calibration in the meantime (see in Figure 4), since the fuzzy KV items in the cache are replaced by precise ones. Consequently, bonus calibration helps to find a better balance between $T_{draft}$ and $\alpha$.

The theoretical analyses above complement the empirical evidence, further supporting that our algorithm can effectively accelerate the inference process.

# F   Limitations

The experiments are conducted on a specific hardware configuration of 8×A100 GPUs and a specific inference system, and the acceleration rate is likely to vary across different platforms due to variations in computation speeds and tensor-parallel efficiency. However, as hardware continues to improve, computational resources may become increasingly redundant, further reducing the benefits of tensor parallelism. Since this issue is going to get worse with faster hardware and operators, we believe that layer parallelism offers a more effective solution by optimizing the utilization of multi-GPU resources now and in future.

# G   Broader impacts

Our work enables faster inference by accelerating speculative decoding in a layer-parallel approach. We believe it has positive broader impact for the area of LLM inference acceleration. To the best of our knowledge, there is no specific negative impacts to be discussed.

Table 9: Detailed results of running time of drafting and verification per 100 tokens, total running time and acceleration ratio, and acceptance rates of two mainstream models across datasets, at temperature=0 and 0.8. The denotation is basically consistent with Table 2.

| Dataset | Method | Llama-3-70B(8B)-Instruct | | | | Qwen2-72B(7B)-Instruct | | | |
|---|---|---|---|---|---|---|---|---|---|
| | | d | v | total | $\alpha$ | d | v | total | $\alpha$ |
| | | | | temperature=0 | | | | | |
| MMLU | vanilla | - | - | 11.61 | - | - | - | 11.88 | - |
| | TP | - | - | 7.57(1.53x) | - | - | - | 7.59(1.56x) | - |
| | +sd | 3.52 | 2.15 | 5.67(2.05x) | 0.57 | 3.32 | 2.25 | 5.57(2.13x) | 0.52 |
| | +tree | 2.52 | 1.59 | 4.11(2.82x) | 0.88 | 2.38 | 1.62 | 4.01(2.96x) | 0.85 |
| | EasySpec | 1.70(↑1.48x) | 1.73 | **3.44(3.38x)** | 0.82 | 1.65(↑1.44x) | 1.70 | **3.35(3.55x)** | 0.80 |
| HE | vanilla | - | - | 11.79 | - | - | - | 11.98 | - |
| | TP | - | - | 7.58(1.55x) | - | - | - | 7.64(1.57x) | - |
| | +sd | 2.93 | 1.79 | 4.72(2.50x) | 0.74 | 2.82 | 1.83 | 4.65(2.58x) | 0.69 |
| | +tree | 2.53 | 1.58 | 4.11(2.87x) | 0.92 | 2.26 | 1.51 | 3.77(3.18x) | 0.95 |
| | EasySpec | 1.61(↑1.57x) | 1.63 | **3.24(3.64x)** | 0.87 | 1.48(↑1.52x) | 1.54 | **3.02(3.97x)** | 0.91 |
| MATH | vanilla | - | - | 11.52 | - | - | - | 11.82 | - |
| | TP | - | - | 7.56(1.52x) | - | - | - | 7.68(1.54x) | - |
| | +sd | 2.96 | 1.74 | 4.70(2.45x) | 0.73 | 2.48 | 1.65 | 4.13(2.86x) | 0.78 |
| | +tree | 2.50 | 1.47 | 3.97(2.90x) | 0.95 | 2.20 | 1.45 | 3.65(3.24x) | 0.96 |
| | EasySpec | 1.58(↑1.58x) | 1.55 | **3.13(3.68x)** | 0.91 | 1.44(↑1.52x) | 1.47 | **2.91(4.06x)** | 0.95 |
| IFEval | vanilla | - | - | 11.29 | - | - | 11.59 | - | - |
| | TP | - | - | 7.55(1.50x) | - | - | - | 7.64(1.52x) | - |
| | +sd | 3.68 | 2.16 | 5.84(1.93x) | 0.55 | 4.24 | 2.80 | 7.04(1.64x) | 0.39 |
| | +tree | 2.53 | 1.55 | 4.09(2.76x) | 0.89 | 2.80 | 1.84 | 4.65(2.49x) | 0.72 |
| | EasySpec | 1.68(↑1.51x) | 1.65 | **3.33(3.39x)** | 0.82 | 1.87(↑1.50x) | 1.94 | **3.81(3.04x)** | 0.67 |
| MGSM | vanilla | - | - | 11.65 | - | - | 11.92 | - | - |
| | TP | - | - | 7.55(1.54x) | - | - | - | 7.63(1.56x) | - |
| | +sd | 2.66 | 1.61 | 4.27(2.73x) | 0.80 | 2.62 | 1.75 | 4.37(2.73x) | 0.72 |
| | +tree | 2.45 | 1.48 | 3.93(2.96x) | 0.96 | 2.12 | 1.50 | 3.62(3.29x) | 0.94 |
| | EasySpec | 1.55(↑1.58x) | 1.51 | **3.06(3.80x)** | 0.93 | 1.54(↑1.37x) | 1.57 | **3.11(3.83x)** | 0.88 |
| | | | | temperature=0.8 | | | | | |
| MMLU | vanilla | - | - | 11.62 | - | - | - | - | - |
| | TP | - | - | 7.59(1.53x) | - | - | - | 7.66(1.56x) | - |
| | +sd | 3.09 | 1.93 | 5.02(2.32x) | 0.67 | 2.90 | 1.93 | 4.82(2.47x) | 0.65 |
| | +tree | 2.49 | 1.54 | 4.02(2.89x) | 0.94 | 2.14 | 1.48 | 3.62(3.30x) | 0.95 |
| | EasySpec | 1.59(↑1.56x) | 1.63 | **3.22(3.61x)** | 0.89 | 1.49(↑1.43x) | 1.54 | **3.03(3.94x)** | 0.93 |
| HE | vanilla | - | - | 11.79 | - | - | - | 12.14 | - |
| | TP | - | - | 7.64(1.54x) | - | - | - | 7.65(1.59x) | - |
| | +sd | 2.94 | 1.81 | 4.75(2.48x) | 0.73 | 2.54 | 1.63 | 4.17(2.91x) | 0.80 |
| | +tree | 2.47 | 1.49 | 3.95(2.98x) | 0.96 | 2.08 | 1.45 | 3.53(3.44x) | 0.97 |
| | EasySpec | 1.53(↑1.62x) | 1.49 | **3.02(3.90x)** | 0.94 | 1.45(↑1.44x) | 1.46 | **2.91(4.17x)** | 0.97 |
| MATH | vanilla | - | - | 11.56 | - | - | 11.86 | - | - |
| | TP | - | - | 7.54(1.53x) | - | - | - | 7.63(1.55x) | - |
| | +sd | 2.81 | 1.65 | 4.46(2.59x) | 0.78 | 2.44 | 1.54 | 3.98(2.98x) | 0.85 |
| | +tree | 2.44 | 1.45 | 3.89(2.97x) | 0.97 | 2.09 | 1.40 | 3.50(3.39x) | 0.99 |
| | EasySpec | 1.52(↑1.61x) | 1.48 | **3.00(3.86x)** | 0.94 | 1.43(↑1.47x) | 1.44 | **2.87(4.13x)** | 0.98 |
| IFEval | vanilla | - | - | 11.30 | - | - | 11.63 | - | - |
| | TP | - | - | 7.53(1.50x) | - | - | - | 7.72(1.51x) | - |
| | +sd | 3.08 | 1.87 | 4.94(2.29x) | 0.67 | 3.24 | 2.14 | 5.38(2.16x) | 0.56 |
| | +tree | 2.45 | 1.46 | 3.91(2.89x) | 0.97 | 2.17 | 1.51 | 3.68(3.16x) | 0.91 |
| | EasySpec | 1.52(↑1.61x) | 1.50 | **3.03(3.73x)** | 0.92 | 1.53(↑1.42x) | 1.58 | **3.11(3.73x)** | 0.87 |
| MGSM | vanilla | - | - | 11.67 | - | - | 12.02 | - | - |
| | TP | - | - | 7.55(1.55x) | - | - | - | 7.65(1.57x) | - |
| | +sd | 2.54 | 1.55 | 4.10(2.85x) | 0.84 | 2.68 | 1.72 | 4.41(2.73x) | 0.76 |
| | +tree | 2.30 | 1.44 | 3.73(3.13x) | 0.98 | 2.08 | 1.45 | 3.53(3.41x) | 0.97 |
| | EasySpec | 1.52(↑1.51x) | 1.49 | **3.01(3.88x)** | 0.96 | 1.46(↑1.42x) | 1.50 | **2.96(4.06x)** | 0.94 |

