# OpenReview forum: "EasySpec: Layer-Parallel Speculative Decoding for Efficient Multi-GPU Utilization"
_NeurIPS.cc/2025/Conference — NeurIPS 2025 poster_

### Official Review · Reviewer_Pf1A · 2025-06-30

**Clarity:** 3
**Significance:** 2
**Originality:** 3
**Rating:** 4
**Confidence:** 5

**Summary:**

This paper introduces EasySpec, a training-free, plug-in strategy for accelerating speculative decoding in multi-GPU systems. By breaking inter-layer dependencies, EasySpec enables parallel execution of draft model layers to maximize GPU utilization. It also calibrates KVcache errors with minimal overhead, and experiments demonstrate its effectiveness.

**Questions:**

The minor concerns are as follows:

1. In Figure 1(b), EasySpec appears to achieve balanced GPU utilization. However, my understanding is that only the attention layers are distributed across GPUs, while the MLP layers remain sequentially executed on a single GPU — is this correct? Additionally, the attention layers do not seem to be evenly distributed across GPUs. For example, in Table 4 (N=4), Strategy=0 uses only one GPU while the other three are idle, and Strategies=1–3 utilize only three GPUs, leaving one unused. This seems inconsistent with the balanced load shown in Figure 1(b), which is somewhat confusing.

2. Table 2 shows that EasySpec outperforms EAGLE-2 in throughput. Is EAGLE-2 using tensor-parallel (TP) decoding in this comparison? Also, are both methods evaluated under the same hardware conditions?

3. During the drafting stage, given that multiple GPUs are available, is it feasible to use multiple distinct draft models simultaneously to generate draft candidates? For example, could both LLaMA-3-8B and Qwen2-7B be used in parallel to accelerate LLaMA-3-70B inference?

**Suggestions**

There are many concurrent works on speculative decoding (e.g., [1–8]); please expand the Related Work section to include more of these, which will help readers better understand the development of this field and EasySpec's unique contributions.

[1] LayerSkip: Enabling Early Exit Inference and Self-Speculative Decoding

[2] SpecExec: Massively Parallel Speculative Decoding for Interactive LLM Inference on Consumer Devices​

[3] Generation Meets Verification: Accelerating Large Language Model Inference with Smart Parallel Auto-Correct Decoding

[4] TriForce: Lossless Acceleration of Long Sequence Generation with Hierarchical Speculative Decoding

[5] EdgeLLM: Fast On-Device LLM Inference With Speculative Decoding

[6] A Theoretical Perspective for Speculative Decoding Algorithm

[7] BiTA: Bi-Directional Tuning for Lossless Acceleration in Large Language Models

[8] OPT-Tree: Speculative Decoding with Adaptive Draft Tree Structure

**Ethical Concerns:**

["NO or VERY MINOR ethics concerns only"]

**Final Justification:**

The response addresses the concerns I raised, and I find the work overall both meaningful and promising.

**Limitations:**

Yes.

**Paper Formatting Concerns:**

N/A.

**Quality:**

3

**Strengths And Weaknesses:**

**Strengths**

1. This paper presents a meaningful attempt to explore speculative decoding in multi-GPU settings.

2. The addressed problem is important and highly relevant for real-world LLM applications.

3. The idea of fuzzy speculation is intriguing; while it may appear methodologically imperfect due to potential performance degradation, the paper convincingly argues that "the drafting result is never required to be precise," and experimental results confirm its limited negative impact.

4. The paper is well-organized and clearly written, making it easy to follow.


**Weakness**

The main concerns are as follows:

1. The layer-parallel (LP) configuration is currently based on empirical heuristics, lacking a systematic approach for partitioning attention layer sets. While the method is training-free, achieving optimal performance requires extensive manual tuning (as shown in Table 4). It would be valuable to explore automated strategies for configuring distributed attention layers.

2. The applicability of EasySpec is somewhat limited, as it depends on the availability of a suitable draft model. This constraint makes it suitable mainly for LLMs where a compatible drafter is readily available (e.g., LLaMA-3-8B for LLaMA-3-70B, Qwen2-7B for Qwen2-72B). For other LLMs, significant output distribution gaps may exist. The paper reports results only on LLaMA-3-70B-Instruct and Qwen2-72B-Instruct. Including evaluations on additional models—such as using LLaMA-3-8B-Instruct to accelerate Vicuna-33B, LLaMA-2-70B, or Qwen2-72B-Instruct—would greatly enhance the paper's value.

3. Key baseline comparisons are missing. While the paper notes that integrating a draft model into a distributed system is non-trivial, tensor-parallel (TP) inference for the draft model is theoretically feasible. Even though Table 6 shows limited speedup for TP > 1, including a direct comparison between a TP-enabled draft model and EasySpec would improve the experimental rigor.

4. Further issues are discussed in the "Questions" section.

Overall, despite these concerns, I find the paper refreshing and creative. My current score falls between borderline reject and borderline accept, but I am open to raising it following constructive author feedback.

---

> ### Author Rebuttal · Authors · 2025-07-30
>
> We sincerely thank you for the review!
>
> We appreciate your positive feedback on the originality, quality and clarity of our paper.
>
> >While the method is training-free, achieving optimal performance requires extensive manual tuning (as shown in Table 4). It would be valuable to explore automated strategies for configuring distributed attention layers.
>
> Thanks for your suggestion! In fact, the strategy is not manually tuned and heuristic, and it is quite easy to follow. For the layer-parallel size N, we parallelize 1 ~ N, N ~ 2N-1, ..., until the remaining layers (except for the last) are not enough to compose a N-set. For example, for N=4, we have 0, 1 ~ 4, 4 ~ 7, ..., 28 ~ 30 parallelized, while layer 28,29, and 30 have to compose a 3-set because layer 31 is the last. We will add such explanation to the paper.
>
> The reason of such strategy is that, we observe the first and last layers should not be parallelized with other layers, and parallelizing other intermediate layers yields the best performance. We hypothesize that this is because the first and last layers are more sensitive to approximation errors.
>
> As stated in appendix B.1, we are not sure wheter this strategy is the best, but it already has sufficiently satisfactory results. Moreover, simply trying other combinations is lightweight, as it requires no modifications to the original model architecture. A change in configuration is enough.
>
> >The paper reports results only on LLaMA-3-70B-Instruct and Qwen2-72B-Instruct. Including evaluations on additional models—such as using LLaMA-3-8B-Instruct to accelerate Vicuna-33B, LLaMA-2-70B, or Qwen2-72B-Instruct—would greatly enhance the paper's value.
>
> Speculative decoding with heterogeneous models is not feasible, as it requires both models to share the same tokenizer. Specifically, token IDs must be identical across models, so cross-family speculation is impossible in principle (e.g., LLaMA to Vicuna or LLaMA-3 to LLaMA-2).
>
> While it is true that EasySpec requires access to a smaller drafter model from the same family, we think that our method still offers great adaptability. In practice, smaller variants are commonly available in major open-source families like LLaMA and Qwen. In contrast, training a new speculation model from scratch is resource-intensive and non-trivial, requiring data, infrastructure, and tuning expertise. Trained drafter can also have unstable performance due to overfitting, as we show that the variance of throughput is much higher.
>
> Lastly, the smaller model does not have to be the exact same version as the larger one, as long as they share the same tokenizer. For example, Llama-3.2-1B/3B can be used to accelerate Llama-3-70B, as we already experimented on. This largely mitigate the alleged limitation.
>
> >While the paper notes that integrating a draft model into a distributed system is non-trivial, tensor-parallel (TP) inference for the draft model is theoretically feasible. Even though Table 6 shows limited speedup for TP > 1, including a direct comparison between a TP-enabled draft model and EasySpec would improve the experimental rigor.
>
> As shown in Table 6, small models cannot benefit from TP>1; in fact, they often experience a performance drop compared to TP=1. This is not suprising, as small models are not computation-bound enough to benefit from tensor parallelism, and the additional communication overhead leads to inefficiencies. Consequently, the overall performance with TP>1 for drafter is often worse than TP=1,and even more worse than EasySpec.
>
> To support this observation, we conducted experiments with TP=4 under various base-draft settings, and the results are presented below:
>
> | Base-Draft | MMLU | IFEval | mgsm |
> | ----  | ----  | ----  | ----  |
> | TP=4 w/o EasySpec | | | |
> | 70B-3B | 22.96 | 21.89 | 23.50 |
> | 70B-1B | 29.25 | 27.02 | 29.88 |
> | 8B-1B | 43.38 | 41.56 | 47.75 |
> | TP=1 w/o EasySpec | | | |
> | 70B-3B | 23.06 | 22.33 | 23.81 |
> | 70B-1B | 29.46 | 27.44 | 30.80 |
> | 8B-1B | 45.12 | 42.83 | 49.90 |
> | EasySpec | | | |
> | 70B-3B | 29.72 | 26.95 | 29.54 |
> | 70B-1B | 32.01 | 29.22 | 33.92 |
> | 8B-1B | 52.23 | 50.45 | 53.88 |
>
> These results clearly demonstrate that EasySpec consistently outperforms both TP = 1 and TP > 1 baselines, reinforcing our claim that layer-parallel EasySpec is better suited than standard TP-based approaches.
>
> >My understanding is that only the attention layers are distributed across GPUs, while the MLP layers remain sequentially executed on a single GPU — is this correct? Additionally, the attention layers do not seem to be evenly distributed across GPUs. For example, in Table 4 (N=4), Strategy=0 uses only one GPU while the other three are idle, and Strategies=1–3 utilize only three GPUs, leaving one unused. This seems inconsistent with the balanced load shown in Figure 1(b), which is somewhat confusing.
>
> Your understanding of MLP sequential running is correct.
>
> Figure 1(b) illustrates the most typical configuration of layer parallelism in our approach, which is N=4 for intermediate layers. Special cases involving the first and last layers are not shown. We will add some extra explanation for it.
>
> Due to the first and last layer not parallelized, EasySpec does not achieve full GPU utilization (although it is significantly more efficient than using TP=1). However, our ultimate goal is not maximizing utilization, but achieving optimal performance. Therefore, if a slight under-utilization caused by the first and last layer ultimately results in better performance, it would be an undoubtedly better option.
>
> >Is EAGLE-2 using tensor-parallel (TP) decoding in this comparison? Also, are both methods evaluated under the same hardware conditions?
>
> Yes, we evaluated EAGLE-2 under tensor parallelism (TP=8, the same as EasySpec), and all experiments were conducted under identical hardware and software conditions for fairness of comparison. EAGLE-2 does not natively support TP, so we re-implemented TP support for it by reusing components from our own codebase, without any modification to its core algorithm.
>
> >During the drafting stage, given that multiple GPUs are available, is it feasible to use multiple distinct draft models simultaneously to generate draft candidates? For example, could both LLaMA-3-8B and Qwen2-7B be used in parallel to accelerate LLaMA-3-70B inference?
>
> No, it is impossible to do so, because the tokenizer of Qwen2 models is different from Llama's.
>
> Lastly, we appreciate that you provided us more related work to enhance the contribution of our paper. Thanks to your detailed review!

---

> > ### Comment · Reviewer_Pf1A · 2025-08-03
> >
> > Thank you for preparing a detailed rebuttal, especially for thoroughly addressing all the concerns I raised (some of which, in hindsight, may have been overly critical). I do find the work overall interesting and promising, and I have raised my score accordingly.

---

> > > ### Author Response · Authors · 2025-08-04
> > >
> > > Thank you for reading the rebuttal and giving such positive feedbacks!

---

### Official Review · Reviewer_Ubkp · 2025-06-30

**Clarity:** 2
**Significance:** 1
**Originality:** 2
**Rating:** 4
**Confidence:** 4

**Summary:**

Conventional speculative decoding has low multi-gpu utilization when the draft model is small. The paper considered the sequential multi-layer inference pipeline as the main bottleneck and proposed a “fuzzy” speculation method, EasySpec, that runs multiple model layers in parallel. To avoid accumulated KV cache errors and potential performance drop, EasySpec introduced Bonus Calibration that recomputed the KV cache using the accepted tokens after each draft-verification cycle. The performance of EasySpec has been evaluated across different LLMs, e.g. LlaMA-3-70B (8B), Qwen2-72B(7B), etc, and different benchmarks, e.g. MMLU, Spec-Bench, etc. Experiments showed that EasySpec achieved lower inference latency compared to several baseline approaches, e.g. TP, tree-attention, etc.

**Questions:**

N.A.

**Ethical Concerns:**

["NO or VERY MINOR ethics concerns only"]

**Final Justification:**

The reviewer appreciated the authors' responses and clarification that resolved the reviewer's concerns. The reviewer raised the rating the paper accordingly.

**Limitations:**

Limitations were included in the Appendix

**Paper Formatting Concerns:**

N.A.

**Quality:**

2

**Strengths And Weaknesses:**

---
Strengths

Method: EasySpec proposed to “break” the sequential dependency of the hidden states across consecutive blocks/layers to a certain extent, which is technically sound. Prior research in LLMs has indicated that the hidden states in consecutive layers may exhibit limited changes in several scenarios, given the use of skip connections [1]

Experiment: the paper provided extensive LATENCY evaluations on auto-regressive inference, including comparisons to alternative approaches on various benchmarks.

---

Weaknesses

Accuracy: it was indicated in the paper that EasySpec may lead to 7% accuracy drop, but no further discussions were included in the paper. This is a main concern. From the paper, EasySpec is not lossless. While EasySpec demonstrated superior throughput, it is not clear to the reader if EasySpec achieved at least comparable accuracies to the baselines. The authors are encouraged to include accuracy comparisons to the baselines across different benchmarks, similar to the latency evaluations. Ablations on the tradeoff between the speedup and the accuracy drop are also highly recommended, as 7% accuracy drop is high.


Baselines: the reader may expect at least one representative TP baseline. However, from L192, it looks like the baseline drafter did not use TP (i.e. TP size == 1). From Table 6, the optimal TP size was determined on MMLU. The reviewer has two questions: i) since MMLU involves only classification tasks that requires no autoregressive generation, the reviewer wondered if the optimal TP sized was determined under the autoregressive generation setting or prefilling setting? ii) Any chance the optimal TP size varies across benchmarks?

Presentation: several figures/tables were referred in the main paper but presented in the appendix, e.g. Figure4, Table 6. The authors are encouraged to reposition the figures and tables


[1] The Unreasonable Ineffectiveness of the Deeper Layers. ICLR 2025.

---

> ### Author Rebuttal · Authors · 2025-07-30
>
> We sincerely thank you for the review.
>
> >It was indicated in the paper that EasySpec may lead to 7% accuracy drop, but no further discussions were included in the paper. This is a main concern. From the paper, EasySpec is not lossless. While EasySpec demonstrated superior throughput, it is not clear to the reader if EasySpec achieved at least comparable accuracies to the baselines. The authors are encouraged to include accuracy comparisons to the baselines across different benchmarks, similar to the latency evaluations. Ablations on the tradeoff between the speedup and the accuracy drop are also highly recommended, as 7% accuracy drop is high.
>
> Firstly, there is a huge misunderstanding here. The term 'accuracy' in speculative decoding are referring to the acceptance rate of speculated tokens—rather than the task-level accuracy. Speculative decoding is inherently a lossless inference acceleration technique, which has been both theoretically and empirically verified in many prior works. The decrease in speculation acceptance rate has no impact on the output quality.
>
> Secondly, we are trading a small drop in speculation accuracy not for overall better performance, but for faster drafting stage. Since the final output quality is not affected, the drop in speculation accuracy is completely reasonable if the drafting stage becomes faster. We believe the effectiveness of this trade-off has been thoroughly demonstrated and justified by our experimental results. Moreover, as a lossless inference acceleration method, the acceptance rate itself is not the ultimate goal, and the overall speed is. The only metric of a speculative decoding method should be the overall performance, which EasySpec achieve higher than previous TP or non-TP baselines.
>
> >The reader may expect at least one representative TP baseline. However, from L192, it looks like the baseline drafter did not use TP (i.e. TP size == 1).
>
> We conducted extensive experiments with TP=4, and it demonstrates that TP>1 is harmful for the drafter and hence the overall speed. EaspSpec is indeed targeted at solving this problem.
>
> | Base-Draft | MMLU | IFEval | mgsm |
> | ----  | ----  | ----  | ----  |
> | TP=4 w/o EasySpec | | | |
> | 70B-3B | 22.96 | 21.89 | 23.50 |
> | 70B-1B | 29.25 | 27.02 | 29.88 |
> | 8B-1B | 43.38 | 41.56 | 47.75 |
> | TP=1 w/o EasySpec | | | |
> | 70B-3B | 23.06 | 22.33 | 23.81 |
> | 70B-1B | 29.46 | 27.44 | 30.80 |
> | 8B-1B | 45.12 | 42.83 | 49.90 |
> | EasySpec | | | |
> | 70B-3B | 29.72 | 26.95 | 29.54 |
> | 70B-1B | 32.01 | 29.22 | 33.92 |
> | 8B-1B | 52.23 | 50.45 | 53.88 |
>
> >Since MMLU involves only classification tasks that requires no autoregressive generation, the reviewer wondered if the optimal TP sized was determined under the autoregressive generation setting or prefilling setting?
>
> In all experiments, including MMLU, we adopt chain-of-thought (CoT) inferece, which naturally results in multi-token autoregressive generation, not single-token outputs. The generation length is restricted to no more than 128. There should be no doubt about this experimental setting: speculative decoding is about a sequence of tokens to be generated and verified, so the experiments can never be purely 1-token generation.
>
> >Any chance the optimal TP size varies across benchmarks?
>
> The optimal tensor parallelism (TP) size may vary across different tasks. However, in our experiments, this influence is negligible. For auto-regressive generation tasks, the prefilling stage typically accounts for only a small portion of inference time, with the majority dominated by the decoding stage. Decoding is basically a 1-token process, so it has similar computation-memory ratio across tasks, and the optimal TP size is also similar.

---

> ### Author Response · Authors · 2025-08-08
> **Reminding response to rebuttal**
>
> Dear Reviewer Ubkp,
>
> We hope this message finds you well. As the discussion period is nearing its end (with less than 2 days remaining), we are writing to kindly remind you to give response to our rebuttal and participate in the discussion.
>
> Thank you for reviewing our paper!

---

### Official Review · Reviewer_RLfo · 2025-07-02

**Clarity:** 2
**Significance:** 3
**Originality:** 2
**Rating:** 4
**Confidence:** 3

**Summary:**

This paper proposes EasySpec, a layer-parallel speculative decoding method designed to accelerate the drafting stage in large language model inference. In conventional speculative decoding, the draft model is typically small and cannot fully utilize multi-GPU resources. EasySpec addresses this by breaking the sequential dependency between attention layers in the draft model, allowing them to run in parallel across multiple GPUs. To mitigate the KV cache errors introduced by this fuzzy execution, it introduces bonus calibration. Experiments are conducted on several large models and tasks, measuring drafting and verification time, overall speedup, and acceptance rate.

**Questions:**

1. While bonus calibration is described as lightweight, is there any scenario in which the cost of re-running the draft model outweighs the benefits of layer-parallelism? Some further clarification would be useful.
2. In fuzzy speculation, the attention layers are executed in parallel using the same input, which generates a set of KV caches. It remains somewhat unclear whether these fuzzy KV caches are meaningfully used in subsequent token generation before being replaced.
3. EasySpec shows strong performance with medium-sized draft models. However, could it extend to significantly smaller drafters?
4. PEARL [1] also focuses on accelerating the drafting stage and overlaps the drafting and verification stages to achieve stage-level parallelism. It would be helpful if the key differences between the two approaches could be clarified.

[1] Liu, T., Li, Y., Lv, Q., Liu, K., Zhu, J., Hu, W., & Sun, X. (2025). Pearl: Parallel speculative decoding with adaptive draft length. In The Thirteenth International Conference on Learning Representations.

**Ethical Concerns:**

["NO or VERY MINOR ethics concerns only"]

**Final Justification:**

Many thanks to the authors for their extensive response and great effort. I feel more positive about this work and would like to upgrade my score.

**Limitations:**

Yes

**Paper Formatting Concerns:**

No.

**Quality:**

2

**Strengths And Weaknesses:**

Strengths:

+ The paper focuses on a practical performance bottleneck in current large language model inference pipelines—the under-utilization of the drafting stage in multi-GPU settings, making the topic of practical importance.
+ The paper is well-organized and easy to follow. Background concepts such as speculative decoding and tensor parallelism are explained with sufficient clarity. The method is described clearly, aided by formulas, algorithm pseudocode, and illustrations.

Weaknesses:

- While the paper identifies the drafting stage as a performance bottleneck, the motivation could be communicated more compellingly. The Introduction currently lacks concrete system-level profiling, visual illustrations, or quantitative comparisons that would help emphasize the severity and relevance of the problem.
- Table 3 presents cosine similarities of intermediate hidden states to support the approximation quality of fuzzy speculation. It would be helpful to include output-level comparisons (e.g., token distributions or generation divergence) to complement the current analysis.
- Figure 1 illustrates the core idea of layer-parallel attention execution but does not provide a full depiction of the entire EasySpec workflow. In particular, bonus calibration and KV cache correction are critical components of the method and would benefit from visual inclusion in a complete process diagram.

---

> ### Author Rebuttal · Authors · 2025-07-30
>
> We sincerely thank you and the ACs for your time and effort.
>
> >While bonus calibration is described as lightweight, is there any scenario in which the cost of re-running the draft model outweighs the benefits of layer-parallelism? Some further clarification would be useful.
>
> Thanks for your suggestion! During our experiments, we did not observe such a scenario in any of our experiments. As shown in our ablation study, bonus calibration consistently improves performance under all tested conditions. This is because, without calibration, the fuzzy errors will accumulate as the generation length increases, eventually leading to a significant drop in accuracy. Our setting of generation length is averagely 128, so this degradation is particularly harmful to overall performance. Consequently, the computational cost of bonus calibration is much outweighed by the gains in speed and accuracy, making it a consistently valuable component of our approach.
>
> >In fuzzy speculation, the attention layers are executed in parallel using the same input, which generates a set of KV caches. It remains somewhat unclear whether these fuzzy KV caches are meaningfully used in subsequent token generation before being replaced.
>
> As shown in Figure 2, the fuzzy KV caches are used only during a single speculation phase, where they serve to generate N speculated tokens (N = 5 in our experiments). Once these tokens are passed to the verification model, the acceptance decisions are returned, and the corresponding calibrated KV cache entries are generated to replace the fuzzy ones. Therefore, the impact of cache fuzziness is strictly limited to a single drafting stage, and does not propagate beyond that phase.
>
> >EasySpec shows strong performance with medium-sized draft models. However, could it extend to significantly smaller drafters?
>
> The results on smaller drafters, such as 0.5B, 1B, 1.5B or 3B model, are in Table 8 of the appendix. As shown, EasySpec is effective in all the drafter-base model combinations across various tasks.
>
> >PEARL also focuses on accelerating the drafting stage and overlaps the drafting and verification stages to achieve stage-level parallelism. It would be helpful if the key differences between the two approaches could be clarified.
>
> In fact, EasySpec and PEARL are fundamentally orthogonal in design. Speculative decoding consists of two stages: drafting and verification. PEARL focuses on inter-stage acceleration, proposing strategies to overlap these two stages for multi-GPU efficiency—either by initiating verification before all draft tokens are generated, or by continuing drafting before verification results are returned. In contrast, EasySpec targets at intra-stage acceleration. It accelerates the drafting stage alone, using fuzzy speculation and then calibration. Thus, EasySpec and PEARL can be seen as complementary approaches that optimize different aspects of the speculative decoding pipeline. We will cite PEARL in the paper for more clarification.
>
>
> >Table 3 presents cosine similarities of intermediate hidden states to support the approximation quality of fuzzy speculation. It would be helpful to include output-level comparisons (e.g., token distributions or generation divergence) to complement the current analysis.
>
> The cosine similarity provides a model-internal perspective on the behavior of EasySpec. For output-level evaluation, we think the most compelling metric is the acceptance accuracy, which directly reflects how well the fuzzy speculation approximates the precise generation. As noted, the speculation accuracy decreases by only 7% at most, and in certain cases—such as code and math tasks—the accuracy drop is even 0. This indicates that layer-parallel fuzzy speculation can closely approximate the true token distribution.
>
> We give further clarification about why cosine similarity is a good metric: for input hidden states $h_1$ and $h_2$, as there is a normalization layer at the entry of the attention module, $h_1$ and $h_2$ become normalized vectors $h'_1$ and $h'_2$, which are input to the subsequent matrices. It is easy to see that cos($h_1$, $h_2$) = cos($h'_1$, $h'_2$), so the cosine similarity between $h_1$ and $h_2$ is indeed reflecting the approximation precision of modules' input.
>
> >Figure 1 illustrates the core idea of layer-parallel attention execution but does not provide a full depiction of the entire EasySpec workflow. In particular, bonus calibration and KV cache correction are critical components of the method and would benefit from visual inclusion in a complete process diagram.
>
> That is a very useful suggestion! We did improve Figure 1 and some other figures to strenthen the clarity, but we cannot link or post them here. It will be put into further versions.

---

> > ### Comment · Reviewer_RLfo · 2025-08-07
> >
> > I appreciate the authors' careful rebuttal, which addresses many of my concerns. I feel much more positive about this paper. After confirming some issues with other reviewers during the subsequent discussion phase, I am happy to upgrade my rating if needed.

---

> > > ### Author Response · Authors · 2025-08-07
> > >
> > > Thank you for reading the rebuttal and giving positive response!
> > >
> > > It is good to know that our clarifications have addressed your concerns.
> > >
> > > If there is any further clarification that can be done by us, please feel free to let us know. We are eager to provide additional explanation or detail about the paper.

---

### Official Review · Reviewer_d76v · 2025-07-03

**Clarity:** 3
**Significance:** 2
**Originality:** 2
**Rating:** 4
**Confidence:** 4

**Summary:**

This paper proposes a more parallelized speculative decoding method, made by modifying draft models to run multiple layers in parallel. They demonstrate that this leads to a real improvement in distributed systems compared to distributed normal specdec. They compare against competitive baselines like Eagle 2, with detailed benchmarking of the time spent in drafting and verification.

**Questions:**

How costly is training new draft heads for Eagle 2? Could you generate draft heads for Llama 3.3, and compare?

How well do distilled models perform as draft models in this use case? Would it be possible to use one of the Llama models trained with distillation to see if it gives superior draft performance?

**Ethical Concerns:**

["NO or VERY MINOR ethics concerns only"]

**Final Justification:**

New experiments with 1B draft models noticeably increased the throughput. Before, the paper was a borderline rejection due to being a highly empirical method without clear empirical superiority. With the improved results, I am not only convinced the method is more practically useful but that it represents an interesting avenue for future research.

**Limitations:**

Yes

**Quality:**

3

**Strengths And Weaknesses:**

Strengths:

Their method is interesting. Its a essentially "free" modification of existing inference pipelines to enable faster decoding. There are definite applications in cases where you do not want to fine tune an additional model. Furthermore, it addresses a real problem with draft model based approaches to speculative decoding, which is their poor utilization of hardware.

Neutral:

There is an existing literature in parallelized LLMs (see, for example, ladder transformers and diffusion LLMs)

Weaknesses:
I do think that the Llama 3.3 comparison against Eagle 2 listed here is unfair. Eagle 2 achieves better performance than EasySpec given a draft model in the same family and trained heads, as shown in the Llama 3 comparison. When adapting to llama 3.3, the performance of easyspec is better, but Eagle 2's draft heads are much easier to train than an entire smaller draft model.

Furthermore, the cost of this type of method is that two different KV caches need to be maintained, which increases memory usage.

---

> ### Author Rebuttal · Authors · 2025-07-30
>
> We sincerely thank you for the review!
>
> > I do think that the Llama 3.3 comparison against Eagle 2 listed here is unfair. Eagle 2 achieves better performance than EasySpec given a draft model in the same family and trained heads, as shown in the Llama 3 comparison. When adapting to llama 3.3, the performance of easyspec is better, but Eagle 2's draft heads are much easier to train than an entire smaller draft model.
>
> We believe that your concern is whether EasySpec can outperform existing state-of-the-art (SOTA) methods like EAGLE-2 on already-trained models LLaMA-3-70B, rather than LLaMA-3.3.
>
> In Section 4.1.2, our goal is to show that, EasySpec can improve the performance of non-trained 8B drafter to 1B trained EAGLE-2 head. To emphasize the gap of drafter sizes (8B vs 1B), we choose the 8B model as drafter, which is quite larger than 3B or 1B one, and hence the performance could also be inferior (but comparable enough to EAGLE-2 already).
>
> To address your concern regarding SOTA performance, we conducted additional experiments using LLaMA-3.2-1B as the drafter on SpecBench:
>
> | | mt-bench | translation | summarization | question-answering | math-reasoning | rag |
> | ---- | ----  | ----  | ----  | ----  | ---- | ---- |
> | 1B w/o EasySpec |33.83 | 30.36 | 33.01 | 31.39 | 37.16 | 33.74 |
> | EAGLE-2 |33.81 | 30.44 | 33.26 | 30.98 | 34.37 | 33.7 |
> | 1B w/ EasySpec |35.15 | 35.03 | 36.11 | 36.12 | 39.17 | 36.32 |
>
> Without EasySpec, the 1B drafter only achieves comparable performance to EAGLE-2, and with EasySpec it clearly outperforms EAGLE-2 across all tasks. This confirms that EasySpec is effective in improving SOTA average throughput as well. Both methods are evaluated under the same hardware and software conditions, for fairness of comparison.
>
> >How well do distilled models perform as draft models in this use case? Would it be possible to use one of the Llama models trained with distillation to see if it gives superior draft performance?
>
> First of all, the smaller model in the same series can be treated as a 'distilled' model indeed. It is a 'distilled' version of the larger, since they share the same training dataset and configurations, and therefore have quite aligned behavior.
>
> EasySpec is orthogonal to distillation (at least theoretically). It leverages the intrinsic property of hidden-state similarity across consecutive layers, which is commonly observed in various models. Distilled models often deviate just slightly from the non-trained original, so we believe this phenomenon will preserve.
>
> We also observe that distilled drafters often have unstable performance, as shown in EAGLE-2's experiment. This problem arises from overfitting to specific training datasets, leading to loss of generality.
>
> Furthermore, distillation can only improve the acceptance rate, rather than reducing the runtime of drafting stage. It is demonstrated that the smaller model in the family (as a 'distilled' version upon the vast pre-training corpora) already performs sufficiently well on acceptance rate (e.g., exceeding 0.8), so it is not the bottleneck, but the runtime of drafting stage is. Further distillation may yield marginal improvements while requiring substantial extra effort.
>
> Since EasySpec already improve the performance of non-distilled 1B drafter better than other baselines, we did not conduct any further experiments on whether to distill the model, and it is indeed out of the scope of our method: training-free and plug-in.

---

> > ### Comment · Reviewer_d76v · 2025-08-04
> >
> > Thank you for the detailed response. The strong results on 1B experiments addressed many of my empirical concerns.

---

> > > ### Author Response · Authors · 2025-08-05
> > >
> > > Thank you for reading the rebuttal!
> > >
> > > We appreciate your suggestion upon the extensive experiment, and it is good to know that our rebuttal addressed the primary concerns.
> > >
> > > Please feel free to let us know if you have any further questions!
> > >
> > > Thanks!

---

### Note · Authors · 2025-08-12

Dear reviewers, AC and SAC,

Thank you for handling our submission!

We greatly appreciate the reviewers’ thoughtful comments and suggestions, and it is good to know that our rebuttal has addressed the main concerns, with no further issues raised. We sincerely thank you for reading our rebuttal and providing feedback.

Before the discussion phase concludes, we would like to note a few points for consideration in the decision-making process:

### 1. Follow-up for Reviewer RLfo

Reviewer RLfo mentioned that some issues would need to be confirmed from other reviewers. We respectfully inquire the other reviewers or AC to share any relevant information that might help confirm these points, and we greatly appreciate your engagement in this discussion.

### 2. Clarification of clear factual errors

We believe that certain statements in Reviewer Ubkp’s review contain clear factual errors.

A. The main concern of Reviewer Ubkp that 'our method is not lossless' is fundamentally incorrect; speculative decoding is both theoretically proven and empirically verified to be lossless in many existing works.

B. The statement that 'the benchmark involves no auto-regressive generation' is also conceptually impossible, as speculative decoding is inherently applied to token-sequence generation and therefore must be auto-regressive. We use CoT generation for all tasks to ensure that.

Despite these errors are clear and obvious and we have clarified them in the rebuttal, we would appreciate it if other reviewers could further help ensure that these aspects are correctly understood.

### 3. Reviewer engagement

We note that Reviewer Ubkp has not responded to our rebuttal or participated in the discussion phase. We believe active participation is important for a fair and appropriate review.

Thank you again for your time and for considering these remarks during the final discussion.

Best regards,

Authors of Paper #6237

---

### Decision · Program_Chairs · 2025-09-17

**Decision:**

Accept (poster)

**Comment:**

This paper proposes EasySpec, a layer-parallel speculative decoding method designed to accelerate the drafting stage in large language model inference.  All reviewers agree that this work makes solid contributions, and the AC recommends acceptance.